# The Transcription Factor EB (TFEB) Sensitizes the Heart to Chronic Pressure Overload

**DOI:** 10.3390/ijms23115943

**Published:** 2022-05-25

**Authors:** Sebastian Wundersitz, Cristina Pablo Tortola, Sibylle Schmidt, Ramon Oliveira Vidal, Melanie Kny, Alexander Hahn, Lukas Zanders, Hugo A. Katus, Sascha Sauer, Christian Butter, Friedrich C. Luft, Oliver J. Müller, Jens Fielitz

**Affiliations:** 1Experimental and Clinical Research Center (ECRC), Max Delbrück Center (MDC) for Molecular Medicine in the Helmholtz Association, Charité-Universitätsmedizin Berlin, 13125 Berlin, Germany; sebastian.wundersitz@charite.de (S.W.); cristinapablotortola@gmail.com (C.P.T.); sibylle.schmidt@charite.de (S.S.); melaniekny@web.de (M.K.); alexanderhahn85@gmx.de (A.H.); lukas.zanders@charite.de (L.Z.); luft@charite.de (F.C.L.); 2DZHK (German Center for Cardiovascular Research), Partner Site Berlin, 10785 Berlin, Germany; 3Laboratory of Functional Genomics, Nutrigenomics and Systems Biology, Max Delbrück Center (MDC) for Molecular Medicine in the Helmholtz Association, Berlin Institute for Medical Systems Biology, 10115 Berlin, Germany; ramon.vidal@gmail.com (R.O.V.); sascha.sauer@mdc-berlin.de (S.S.); 4Berlin Institute of Health (BIH), 10178 Berlin, Germany; 5Department of Internal Medicine III, University Hospital Heidelberg, 69120 Heidelberg, Germany; hugo.katus@med.uni-heidelberg.de; 6DZHK (German Centre for Cardiovascular Research), Partner Site Heidelberg/Mannheim, 69120 Heidelberg, Germany; 7Heart Center Brandenburg, Department of Cardiology, Medical University Brandenburg (MHB), 16321 Bernau, Germany; c.butter@immanuel.de; 8Department of Cardiology and Angiology, University Medical Center Schleswig-Holstein, 14105 Kiel, Germany; oliver.mueller@uksh.de; 9DZHK (German Centre for Cardiovascular Research), Partner Site Hamburg/Kiel/Lübeck, 20251 Hamburg, Germany; 10DZHK (German Center for Cardiovascular Research), Partner Site Greifswald, 17475 Greifswald, Germany; 11Department of Internal Medicine B, Cardiology, University Medicine Greifswald, 17475 Greifswald, Germany

**Keywords:** heart failure, left ventricular hypertrophy, transcription factor EB

## Abstract

The transcription factor EB (TFEB) promotes protein degradation by the autophagy and lysosomal pathway (ALP) and overexpression of TFEB was suggested for the treatment of ALP-related diseases that often affect the heart. However, TFEB-mediated ALP induction may perturb cardiac stress response. We used adeno-associated viral vectors type 9 (AAV9) to overexpress TFEB (AAV9-Tfeb) or Luciferase-control (AAV9-Luc) in cardiomyocytes of 12-week-old male mice. Mice were subjected to transverse aortic constriction (TAC, 27G; AAV9-Luc: *n* = 9; AAV9-Tfeb: *n* = 14) or sham (AAV9-Luc: *n* = 9; AAV9-Tfeb: *n* = 9) surgery for 28 days. Heart morphology, echocardiography, gene expression, and protein levels were monitored. AAV9-Tfeb had no effect on cardiac structure and function in sham animals. TAC resulted in compensated left ventricular hypertrophy in AAV9-Luc mice. AAV9-Tfeb TAC mice showed a reduced LV ejection fraction and increased left ventricular diameters. Morphological, histological, and real-time PCR analyses showed increased heart weights, exaggerated fibrosis, and higher expression of stress markers and remodeling genes in AAV9-Tfeb TAC compared to AAV9-Luc TAC. RNA-sequencing, real-time PCR and Western Blot revealed a stronger ALP activation in the hearts of AAV9-Tfeb TAC mice. Cardiomyocyte-specific TFEB-overexpression promoted ALP gene expression during TAC, which was associated with heart failure. Treatment of ALP-related diseases by overexpression of TFEB warrants careful consideration.

## 1. Introduction

Heart failure with reduced ejection fraction is a common cause of death worldwide and is commonly preceded by pathological cardiac hypertrophy and left ventricular remodeling [1,2]. Pathological hypertrophy is a maladaptive response related to hemodynamic stress stimuli, such as pressure-overload, due to arterial hypertension or aortic stenosis [3]. Although several mechanisms underlying cardiac hypertrophy are well described [4], pathways involved in cardiomyocyte stress response are not well understood. However, disturbed protein homeostasis in cardiomyocytes caused by decreased protein synthesis and increased protein degradation has been implicated [4]. In cardiomyocytes, protein degradation is predominantly mediated by the ubiquitin-proteasome system (UPS) [5,6,7,8] and the autophagy-lysosomal pathway (ALP) [9,10,11]. In contrast to the UPS which degrades single long-lived and misfolded proteins [12], the ALP degrades entire cell contents such as proteins, insoluble protein aggregates, and even whole organelles such as mitochondria [12]. Because the cardiac proteome changes during hypertrophy, activation of protein degradation is important to efficiently remodel the contractile apparatus and assure adaptation of the heart to stress. A malfunction of protein degradation may in turn lead to the accumulation of proteins and pathological stress response. For example, a malfunction of the UPS is associated with hypertrophic cardiomyopathy [13], maladaptive cardiac hypertrophy [5,7] and protein-surplus myopathy [6,8,14]. The ALP is equally important for the maintenance of normal cardiac structure and function and its inhibition in ALP-related diseases often leads to cardiomyopathy [9,15]. For example, in Pompe’s disease, lysosomal acid alpha-glucosidase deficiency compromised the breakdown of glycogen and causes the accumulation of autophagic vesicles and autophagic debris in heart and skeletal muscle leading to heart and skeletal muscle failure [16]. The transcription factor EB (TFEB) is a critical regulator of ALP and is a master regulator of lysosomal biogenesis and autophagy. TFEB activation results in an increased expression of genes involved in autophagy and autophagic flux, biogenesis of autophagosomes, and the fusion of autophagosomes with lysosomes in non-myocytes [17]. TFEB also promotes the expression of genes involved in early and late lysosomal biogenesis as well as UPS-dependent protein degradation [18,19,20,21].

The nodal function of TFEB in ALP promoted the development of TFEB-based gene therapy predominantly by its overexpression to treat lysosomal storage disorders, which was successfully tested in cell and mouse models of Pompe’s disease [22,23], Gaucher’s disease [24], and amyotrophic lateral sclerosis [25]. However, because cardiomyocyte ALP is important for load-induced cardiac stress response [9,10,11,12], its activation by TFEB overexpression may elicit cardiac side effects, especially during hemodynamic stress such as pressure-overload. We therefore tested the hypothesis that TFEB overexpression in cardiomyocytes sensitizes the heart to a pressure-overload-induced cardiac stress response.

## 2. Results

### 2.1. TFEB Overexpression Leads to Heart Failure in Response to Pressure Overload

AAV9-mediated systemic delivery of the murine *Tfeb* cDNA under the control of the cardiomyocyte-specific MLC v2.1 promoter led to an increased cardiac TFEB mRNA expression (Figure 1A) and protein content (Figure 1B). The hearts of sham-treated AAV9-Tfeb and AAV9-Luc mice were similar in size (Figure 1C,D, Appendix A). In transthoracic echocardiography (Figure 2A), AAV9-Tfeb mice showed a slight increase in the left ventricular enddiastolic diameter (LVEDD; *p* < 0.05) and the left ventricular endsystolic diameter (LVESD; *p* < 0.05), and a small decrease in left ventricular ejection fraction (LVEF; *p* < 0.05) (Figure 2B–D, Appendix A) when compared to AAV9-Luc.

Following TAC, AAV9-Luc mice showed an increased heart weight-to-tibia length ratio (HW/TL) to 138% (*p* < 0.001; Figure 1D), a thickening of the interventricular septal wall thickness at systole (IVSths, *p* < 0.05; Appendix A) and diastole (IVSthd, *p* < 0.001; Figure 2F), and an increased thickness of the left ventricular posterior wall at diastole (PWthd, *p* < 0.005; Figure 2G) when compared to AAV9-Luc sham mice. LVESD (*p* < 0.05) was increased whereas the LVEDD remained unchanged in TAC-operated AAV9-Luc mice when compared to sham animals (Figure 2B,C). Although the fractional shortening (FS) was reduced (*p* < 0.01), LVEF and stroke volume (SV) remained unchanged in AAV9-Luc during pressure-overload (Figure 2D,E, Appendix A). Moreover, the lung weight-to-tibia length ratio remained unchanged in AAV9-Luc TAC compared to sham mice (Figure 1E). These data indicate that TAC surgery resulted in compensated left ventricular hypertrophy in AAV9-Luc mice. In contrast, TAC-operated AAV9-Tfeb mice showed a significantly stronger increase in HW/TL ratio when compared to AAV9-Luc mice (AAV9-Tfeb Sham: 100 ± 9%, AAV9-Tfeb TAC: 162.3 ± 19.6%; *p* < 0.001; *p* = 0.014 AAV9-Luc TAC vs. AAV9-Tfeb TAC) (Figure 1D, Appendix A). AAV9-Tfeb TAC mice showed only a mild increase in IVSthd (*p* < 0.01) and PWthd (*p* < 0.01), whereas IVSths, PWths remained unchanged when compared to AAV9-Tfeb sham mice (Figure 2F,G). TAC caused a pronounced increase in LVEDD (*p* < 0.005 vs. AAV9-Tfeb sham) and LVESD (*p* < 0.005 vs. AAV9-Tfeb sham) and a decrease in LVEF (*p* < 0.005 vs. AAV9-Tfeb sham) and FS (*p* < 0.005 vs. AAV9-Tfeb sham) in AAV9-Tfeb compared to sham mice (Figure 2B–E). Importantly, the increase in LVEDD (*p* < 0.001 vs. AAV9-Luc TAC) and LVESD (*p* < 0.001 vs. AAV9-Luc TAC) was substantially stronger and the decrease in LVEF (*p* < 0.001 vs. AAV9-Luc TAC) and FS (*p* < 0.001 vs. AAV9-Luc TAC) was significantly higher in AAV9-Tfeb TAC compared to AAV9-Luc TAC (Figure 2B–E). TAC also caused an increase in E/A-ratio in AAV9-Luc and AAV9-Tfeb; this increase was significantly higher in AAV9-Tfeb mice. The mitral A velocity was significantly lower in TAC-operated AAV9-Tfeb compared to AAV9-Luc mice (Appendix A). Moreover, lung weight-to-tibia length ratio increased in AAV9-Tfeb TAC mice (+222%; *p* < 0.01 vs AAV9-Tfeb sham, *p* < 0.01 vs. AAV9-Luc TAC; Figure 1E) indicative for pulmonary congestion due to heart failure. In summary, in a mouse model of compensated left ventricular hypertrophy cardiomyocyte-specific TFEB overexpression sensitized the heart to chronic pressure-overload resulting in left ventricular dilatation, reduced left ventricular function, and diminished hypertrophic response and diastolic dysfunction with a restrictive filling pattern.

### 2.2. TFEB Overexpression Aggravates Pressure-Overload Induced Cardiac Stress Response

qRT-PCR was used to investigate if the heart failure phenotype in AAV9-Tfeb TAC mice was accompanied by a stronger activation of cardiac stress markers. No differences in *Nppa* (encoding atrial natriuretic factor), *Nppb* (B-type natriuretic peptide), *Myh7* (beta myosin heavy chain; β-MyHC), and *Myh6* (alpha myosin heavy chain, α-MyHC) expression were found between AAV9-Tfeb sham and AAV9-Luc sham mice (Figure 3A–D). When compared to the respective sham group, a significant increase in *Nppa*, *Nppb*, and *Myh7* expression and a significant decrease in *Myh6* expression were found in AAV9-Luc TAC and AAV9-Tfeb TAC mice. Importantly, a significantly higher increase in *Nppa* (*p* < 0.01), *Nppb* (*p* < 0.05) and *Myh7* (*p* < 0.01) expression was found in AAV9-Tfeb TAC compared to AAV9-Luc TAC mice. For *Myh7*, this result was confirmed on the protein level (β-MyHC; Figure 3E). These data show that cardiomyocyte-specific overexpression of TFEB leads to an aggravated cardiac stress response during pressure-overload.

### 2.3. TAC-Induced Interstitial Fibrosis Is Augmented by TFEB Overexpression

Pathological cardiac remodeling is often accompanied by interstitial fibrosis [26,27]. To investigate if the heart failure phenotype resulting from TFEB overexpression was associated with increased interstitial myocardial fibrosis, we performed Picrosirius Red staining of histological cross-sections from our experimental groups. As expected, AAV9-Luc mice developed interstitial fibrosis (Figure 3F,G). However, this phenotype was much stronger in AAV9-Tfeb mice in response to TAC (Figure 3F,G). Although, the gene expression of *connective tissue growth factor* (*Ctgf*), a marker for extracellular matrix synthesis, was higher in the AAV9-Tfeb TAC compared to AAV9-Luc TAC mice, the expression of *collagen I* and *collagen III* remained unchanged (Figure 3H–J). In summary, these data indicate that the heart failure phenotype in AAV9-Tfeb mice is accompanied by pronounced interstitial cardiac fibrosis.

### 2.4. TFEB Increases ALP Gene Expression during Chronic Pressure Overload

The role of TFEB for ALP-mediated protein degradation is well described in non-muscle tissue [17,18,28] but the regulation of ALP-related genes in the heart in response to stress is not well understood. We used next-generation sequencing of total RNA isolated from the interventricular septum of sham- and TAC-operated AAV9-Luc (sham: *n* = 3; TAC: *n* = 4) and AAV9-Tfeb (sham: *n* = 3; TAC: *n* = 3) mice to quantitate TFEB-dependent gene expression. As a result, our data set included 413,676,271 reads sequenced across the four experimental groups in total, where 326,734,645 reads could be uniquely mapped (78.98%) (Appendix A). Principal component analysis (PCA) disclosed pressure-overload as the main source of the variance observed in our dataset (PC1). Furthermore, PC2 separated AAV9-Luc TAC from AAV9-Tfeb TAC mice (Appendix A).

A comparison of the transcriptional profiles revealed that TFEB overexpression caused a differential regulation of only 79 genes (20 up- and 59 downregulated) in the heart (Appendix A). Gene ontology (GO) term analysis (biological process, BP) of these differentially expressed genes (DEG) revealed an enrichment of genes contained in cell activation (GO:0001775; e.g., *Itga2b*, *Itgb2*, *Irs2*, *Ctsl*) and response to stress (GO:0006950; e.g., *Tmsb4x*, *Ppargc1a*, *Gadd45a*). Kyoto Encyclopedia of Genes and Genomes (KEGG)-pathway analysis showed that the regulation of actin cytoskeleton (mmu04810; e.g., *Tmsb4x*, *Itgb2*, *Itga2b*) and phagosome (mmu04145; e.g., *Ctsl*, *Tubb1*, *Itgb2*) was increased in AAV9-Tfeb sham-treated hearts. TAC resulted in an up-regulation of 2405 genes in AAV9-Tfeb but only 1392 genes in AAV9-Luc hearts (Appendix A, Appendix A). From the upregulated genes, 990 genes were increased in both TAC groups. TAC resulted in a downregulation of 2002 genes in AAV9-Tfeb but only 969 genes in AAV9-Luc hearts (Appendix A). From the down-regulated genes, 783 genes were decreased in both TAC groups. These data indicate that the overexpression of TFEB results in an augmented transcriptional regulation in the heart in response to TAC. Upregulated as well as downregulated genes were annotated using the KEGG database to identify enriched categories. Pathways that are likely influenced by DEG are shown in Appendix A. In general, the genes that were increased in both TAC-treated AAV9-Luc and AAV9-Tfeb hearts reflected the general stress response with enrichment of extracellular matrix-receptor interaction (e.g., *Itgb5*, *Fn1*, *Thbs4*, *Thbs3*, *Col1a1*, *Itga5*), focal adhesion (e.g., *Lamc2*, *Rap1b*, *Flnc*, *Cav3*), regulation of actin cytoskeleton (e.g., *Actb*, *Actg1*, *Actr3*, *Actn1*, *Actn4*) and PI3K-Akt signaling pathway (e.g., *Fgf6*, *Myc*, *Pdgfc*, *Spp1*, *Eif4ebp1*, *Creb3*, *Il6*, *Tlr4*, *Atf4*). The genes that were downregulated in both groups were contained in the KEGG pathways fatty acid degradation (e.g., *Acaa2*, *Cpt2*, *Aldh2*, *Hadh*), calcium signaling pathway (e.g., *Camk2a*, *Adrb1*, *Cacna1h*, *Mylk*, *Casq1*), and cardiac muscle contraction (e.g., *Atp2a2*, *Tnni3*, *Myh6*). We next performed a KEGG pathway analysis with the genes that were either upregulated only in the hearts of TAC-treated AAV9-Luc but not AAV9-Tfeb mice and vice versa, which uncovered pronounced differences between both groups. When considering genes that were only upregulated in TAC AAV9-Luc but not TAC AAV9-Tfeb hearts, the NOD-like receptor signaling pathway (e.g., *Casp1*, *Nlrp3*, *Myd88*, *Gsdmd*, *Il1b*), the NF-κB signaling pathway (e.g., *Cd40*, *Prkcb*, *Tnfaip3*) and cytokine–cytokine receptor interaction (e.g., *Ccl12*, *Osm*, *Lif*, *Inhba*, *Il1rl2*, *Ccl2*, *Il7r*) were predominantly enriched. In TAC AAV9-Tfeb hearts, the upregulated genes were contained in the KEGG pathways ribosome (e.g., *Rpl4*, *Rpl10*, *Rplp1*), phagosome (e.g., *Ctss*, *Lamp1*, *Ctsl*, *Atp6v1f*, *Atp6ap1*, *Atp6v0e*), endocytosis (e.g., *Snx4*, *Ldlrap1*, *Snx5*, *Vps37b*, *Chmp4c*, *Chmp3*, *Chmp4b*), autophagy (e.g., *Becn1*, *Gabarapl1*, *Wipi1*, *Mtor*, *Camkk2*, *Atg101*, *Akt1*, *Sqstm1*, *Ctsb*), insulin signaling (e.g., *Ptpn1*, *Irs2*, *Socs2*, *Rapgef1*) and lysosome (e.g., *Atp6ap1*, *Cltb*, *Ctss*, *Lamp1*, *Ctsl*, *Ctsd*, *Ctsb*). Downregulated genes in AAV9-Luc but not AAV9-Tfeb TAC hearts were contained in metabolic pathways (e.g., *Eno3*, *Ldhb*, *Car4*, *Got1*), pyruvate metabolism (e.g., *Adh1*, *Mdh1*, *Acacb*) and oxidative phosphorylation (e.g., *Atp5a1*, *Ndufs2*, *Sdhc*, *Cox6c*). In contrast, downregulated genes in the hearts of TAC-treated AAV9-Tfeb but not AAV9-Luc mice showed enrichment in the cGMP-PKG signaling pathway (e.g., *Pde2a*, *Nfatc3*, *Creb1*, *Pde3a*), growth hormone synthesis, secretion and action (e.g., *Stat5a*, *Stat5b*, *Irs1*, *Mapk14*, *Mapk9*), and adrenergic signaling in cardiomyocytes (e.g., *Camk2b*, *Ryr2*, *Myl2*, *Cacna1s*, *Scn4b*) (Appendix A). GO-term analyses are depicted in Appendix A. To further investigate if genes contained in ALP were increased in TAC treated AAV9-Tfeb hearts when compared to AAV9-Luc, we performed hierarchical clustering of all genes contained in autophagy (Figure 4A) and lysosomes (Figure 4B), respectively. We found that ALP-related genes were increased in both groups in response to TAC. However, this increase was much stronger in TAC-treated AAV9-Tfeb mice (Figure 4).

To confirm that TFEB overexpression increases ALP gene expression in response to chronic pressure-overload, we performed a qRT-PCR analysis of RNAs isolated from the interventricular septum of all experimental groups. In the absence of pressure-overload, AAV9-Tfeb increased the expression of *cathepsin L* (*Ctsl*; *p* < 0.001) and *tectonin beta-propeller repeat containing 1* (*Tecpr1*; *p* < 0.05) (Figure 5). TAC treated AAV9-Luc animals showed increased levels of the ALP-genes *autophagy-related 101* (*Atg101*, *p* < 0.01), *Beclin-1* (*Becn1*, *p* < 0.05), *ATPase H + transporting v1 subunit h* (*Atp6v1h*; *p* < 0.01) and *cathepsin L* (*Ctsl*; *p* < 0.001) (Figure 5A). The induction of *Tecpr1*, *cathepsin D* (*Ctsd*) and *Ctsl* expression was significantly stronger in AAV9-Tfeb TAC when compared to AAV9-Luc TAC mice. In addition, the ALP-genes *gamma-aminobutyric acid receptor-associated protein-like* (*Gabarapl2*; *p* < 0.05), *sequestosome 1* (*Sqstm1*; *p* < 0.05) and *WD repeat domain phosphoinositide-interacting protein 1* (*Wipi1*, *Atg18*; *p* < 0.001), were significantly induced in AAV9-Tfeb TAC mice compared to sham. However, *microtubule-associated protein 1A/1B light chain 3B* (*Map1lc3b*), encoding for LC3B, expression remained unchanged, whereas both isoforms LC3-I and LC3-II increased at the protein level in the two TAC groups, with a stronger increase in AAV9-Tfeb mice (Figure 5B,C). For p62 and Beclin-1 protein contents, we observed an increase in both TAC groups, which was more pronounced in AAV9-Tfeb mice. However, the degree of p62 induction was not consistent for all AAV9-Tfeb TAC animals. In summary, our data show that cardiomyocyte-specific overexpression of TFEB promotes the expression of ALP genes during TAC, which was associated with the occurrence of heart failure in response to pressure-overload.

## 3. Discussion

The function of TFEB in non-muscle cells and tissue is well defined [17,18,28]; however, whether TFEB plays a role in cardiac stress response was unknown. In our study, we used systemic administration of AAV9 to effectively overexpress TFEB in murine cardiomyocytes and exposed the mice to chronic pressure-overload by TAC surgery. Our mouse model caused a compensated left ventricular hypertrophy, as indicated by an increased heart weight, an increased left ventricular wall thickness, a preserved LVEF, and an absence of pulmonary congestion. We found that cardiomyocytes-specific overexpression of TFEB resulted in heart failure as indicated by left ventricular dilatation, reduced systolic function, increased the expression of stress response genes, augmented interstitial fibrosis and pulmonary congestion. Using RNA sequencing, qRT-PCR and Western Blot analysis we show that the overexpression of TFEB was accompanied by increased ALP gene expression in the heart when hemodynamic stress was applied. Our data suggest that TFEB sensitizes the heart to hemodynamic stress and facilitates heart failure possibly due to the activation of ALP-mediated protein degradation. This hypothesis is supported by the notion that TFEB regulates the expression of genes involved in lysosomal biogenesis and phagosome formation, two major parts of the ALP [17,18,19,29,30,31]. Moreover, we observed an increase in ALP gene expression in the heart by cardiomyocyte-specific overexpression of TFEB. Previous studies showed that ALP is important for protein turnover during cardiac stress response, especially in pressure-overload-induced cardiac hypertrophy and heart failure [9,10,11,32,33]. Because cardiomyocytes remodel during pathological hypertrophy with an increased synthesis of sarcomeric and non-sarcomeric proteins, such as β-MyHC or calcium handling proteins, a well-controlled protein degradation must accompany this process to maintain cardiac function. Indeed, hemodynamic stress was shown to activate ALP in cardiomyocytes as early as 24 h after the induction of pressure-overload, which remained elevated for 3 weeks [9] to 8 weeks [34]. Both an increased as well as a decreased ALP activity are associated with cardiac hypertrophy and heart failure [35,36]. For example, chronic pressure-overload-induced heart failure was paralleled by increased autophagy in rodent hearts [9,36]. Likewise, the overexpression of Beclin-1, a key element of autophagy, increased autophagy in response to stress and augmented pathological remodeling in the heart, whereas Beclin-1 haploinsufficiency showed the opposite effect. Moreover, prolonged activation of autophagy by the activation of phosphoinositide 3-kinase accelerated the transition from hypertrophy to heart failure [37]. In contrast, inhibiting autophagy by cardiomyocyte-specific deletion of autophagy-related 5 (Atg5) leads to cardiomyopathy in adult mice [9,15]. Our data support the hypothesis that prolonged ALP activation sensitizes the heart to hemodynamic stress resulting in heart failure. We hypothesize that fine-tuning of the ALP throughout the disease course is important for cardiac stress response.

The transcriptional activity of TFEB is tightly controlled by its phosphorylation, which regulates its subcellular localization [38]. In non-myocytes, TFEB is predominantly localized in the cytoplasm and kinases such as the mammalian target of rapamycin complex 1 (mTORC1), extracellular signal–regulated kinase 1/2 (ERK1/2) and Akt/Protein Kinase B [29,30,39,40,41] phosphorylate TFEB on specific serine residues, which hampers its translocation into the nucleus and its transcriptional activity [17,29,30,39]. For example, activated mTORC1 phosphorylates TFEB at the lysosomal membrane promoting its interaction with 14-3-3 chaperon proteins and therefore its cytosolic retention [30,39]. Conversely, reduced mTORC1 activity enhances TFEB shuttling into the nucleus and increases the expression of ALP genes [29,30,39]. Accordingly, mTORC1 deficiency should increase and its overexpression should decrease TFEB activity. When exposed to TAC *mTorc1* deficient mice showed an increased ALP activity, which was accompanied by left ventricular dilatation and heart failure [42,43]. These data are reminiscent of our findings showing that TFEB overexpression increased ALP gene expression in a mouse model of chronic pressure-overload which was accompanied by heart failure as well. However, whether TFEB activity is increased in the hearts of *mTorc1* deficient mice undergoing TAC and whether this response is accountable for the phenotype observed needs to be proven. In addition, since mTORC1 activity increases during the early [39,40,42,44] and decreases during later time points of TAC when the hearts go into failure [42], we hypothesize that TFEB activity is dynamically regulated throughout the disease course, an idea that warrants further investigations.

Earlier, we showed that the stress-responsive kinase protein kinase D1 (PKD1) via phosphorylation and inactivation of the transcriptional inhibitor histone deacetylase 5 (HDAC5) can also increase TFEB activity [20]. PKD1 activation would therefore be expected to cause a similar response as we observed here. Indeed, cardiomyocyte specific PKD1 overexpression was shown to result in left ventricular dilation, thinning of the ventricular wall, deterioration of cardiac function, and increased expression of cardiac stress markers [45]. This observation is like the heart failure phenotype reported here. Likewise, we have shown that mice with a cardiomyocyte-specific deletion of *Prkd1* (encoding PKD1) are protected from pathological cardiac remodeling and dysfunction induced by stress, such as TAC and isoproterenol-treatment [26]. However, whether the phenotype of PKD1 transgenic mice or *Prkd1* knockout mice is related to a concomitant TFEB activation and inhibition, respectively, needs further investigation. Further work revealed that TFEB and TFE3 activity is also controlled by other PKD family members such as PKD2 and PKD3 and class IIa HDACs including HDAC4 and HDAC7 [21]. If this regulatory network affects TFEB and TFE3-mediated ALP genes in cardiomyocytes warrants further studies.

Without hemodynamic stress, TFEB overexpression caused only minor changes in ALP gene expression. Given the body of evidence showing that TFEB is tightly controlled by stress-responsive kinases [38], we assume that TFEB is less active in the unstressed heart. We hypothesize that in response to pressure-overload, TFEB is then activated, translocates to the nucleus, and increases ALP gene expression. This hypothesis is supported by our RNA sequencing data showing that significantly more ALP genes were increased in AAV9-Tfeb TAC mice, compared to controls. By using qRT-PCR, we confirmed that the expression of some ALP genes was increased in the hearts of TAC-treated AAV9-Tfeb mice. However, the content of ALP proteins, such as *Sqstm1*/p62, was not increased to the same degree in all of the TAC-treated mice. Indeed, p62 itself is a target for ALP-mediated degradation, which would lead to a decrease in its protein content. This balance between gene expression and protein degradation was not equal in all animals, at least not for the 28-day time point at which the analyses were performed. Further studies, especially autophagic flux assays, are needed to investigate the effects of TFEB on ALP activity in cardiomyocytes in response to TAC. Since we have also only investigated male mice, further studies are needed to address sex-based differences in the effects of TFEB-induced ALP activity in heart failure. We will pursue this issue further.

The central role of TFEB in the regulation of ALP activity made the transcription factor an attractive tool to treat lysosomal storage diseases, such as Alzheimer’s disease, amyotrophic lateral sclerosis, Gaucher disease, and Pompe’s disease [22,24,25,46,47,48]. Because Pompe’s disease is a prototypic lysosomal storage disease, it received particular attention [16]. Pompe’s disease is caused by mutations of the GAA gene with resultant acid alpha-glucosidase (GAA, acid maltase) deficiency, a lysosomal hydrolase involved in the breakdown of glycogen. A lack of functional GAA results in extensive intra-lysosomal glycogen storage and accumulation of autophagic vesicles and autophagic debris in muscles [23]. These accumulations result in functional cardiac impairment and skeletal muscle weakness. Gatto et al. [23] elegantly showed that TFEB overexpression for 3 months attenuated the phenotype of Pompe’s disease. As in our study, they also used systemic AAV9 mediated delivery of the human TFEB gene with the difference that they applied the muscle creatine kinase promoter to overexpress TFEB (AAV2.9-MCK-TFEB) in myocytes of the skeletal muscle and the heart. This approach attenuated muscle pathologies in a mouse model of Pompe’s disease and resulted in an improved heart and skeletal muscle structure and function [23]. Together with our data, their results indicate that the overexpression of TFEB per se from 28 days to 3 months does not cause cardiac pathology. However, if the overexpression of TFEB over a longer period could lead to cardiac side effects, especially in mice that do not have a lysosomal storage disease, has not been investigated. There are also some technical differences between the study of Gatto et al. and our work. Gatto et al. overexpressed the human whereas we used the mouse *TFEB* gene. Furthermore, in contrast to the MCK-promoter used by Gatto et al., we used a CMV-enhanced myosin light chain (MLC) v2.1 promoter to favor cardiomyocyte-specific expression and to avoid expression in other cells. These experimental approaches of our respective groups were chosen according to the scientific questions asked. Gatto et al. investigated whether TFEB overexpression would correct Pompe‘s disease in skeletal muscle, whereas we investigated the effects of TFEB in cardiomyocytes in response to pressure overload. Although the overexpression of TFEB seems to be safe under physiological conditions, it might be harmful in response to stress. Our data suggest that TFEB gene transfer needs to be specifically targeted, excluding cardiomyocytes. However, if cardiomyocyte-specific targeting and overexpression of TFEB in cardiomyocytes are needed, for example, to reverse cardiac hypertrophy associated with Pompe’s disease, it may be important to further increase tissue-specific delivery. In this regard, a careful selection of the cell type to be targeted and the use of cell type-specific promoters are necessary. It may also be important to use constructs that can be switched on and off or can be regulated to adjust gene expression and therefore protein amounts if needed. Cell type-specific delivery may also help to improve the efficacy of gene therapy and several delivery methods were shown to be successful. For example, direct intramyocardial injection of AAV was used to deliver the *vascular endothelial growth factor* gene that induced angiogenesis locally in ischemic mouse myocardium [49]. This method is independent of coronary blood flow and is associated with less systemic vector exposure. Another approach is the anterograde infusion of AAV into coronary arteries, which was already practiced in clinical trials of human heart failure patients to restore sarco/endoplasmic reticulum Ca^2+^ ATPase activity [50,51]. However, these trials were not effective in terms of clinical endpoints [51]. Finally, we have previously shown that retrograde infusion of AAV-luciferase vectors loaded to lipid microbubbles into the anterior interventricular coronary vein increased transgene expression in pig hearts. The targeting of cardiomyocytes and gene expression in the heart was further increased when ultrasound-targeted microbubble destruction was used [52].

Based on our data we conclude that if *Tfeb* gene transfer is developed further for the treatment of lysosomal storage diseases, caution is needed for accompanying cardiovascular risk factors, such as arterial hypertension, valvular disorders, and coronary heart disease, and perhaps even in acute stress situations.

## 4. Materials and Methods

### 4.1. Construction of Adeno-Associated Vectors

The generation and administration of adeno-associated vectors (AAV) were performed as recently described [53,54]. Briefly, the AAV9 serotype was used because it shows the highest tropism for rodent hearts when delivered through the tail vein [55,56]. *Tfeb*-cDNA was PCR amplified using the primers shown in Appendix A and cloned into the AAV9-vector genome cassette under the control of the 1.5kb myosin light chain (MLC) 2v promoter and CMV enhancer to increase tissue specificity of AAV-mediated gene transfer (AAV9-Tfeb) [53]. Vectors containing a Luciferase gene served as a control (AAV9-Luc). Viral vector stock production using respective helper plasmids, as well as purification by filtration and iodixanol step gradient centrifugation, dialysis and titration were performed as previously reported [53,54,57].

### 4.2. Animal Model

Animal procedures performed in accordance with the Max-Delbrück Center for Molecular Medicine guidelines were approved by the Landesamt für Gesundheit und Soziales, Berlin, Germany (G 0229/11), and followed the “Principles of Laboratory Animal Care” (NIH publication No. 86-23, revised 1985) and the current version of German Law on the Protection of Animals. Mice were kept on a 12/12-h dark-light cycle (lights on at 7 am) in a temperature (22–24 °C) and humidity controlled (50–60%) environment and provided ad libitum standard chow and water. To induce chronic pressure-overload, 12-week-old male C57BL/6J wild type mice were subjected to transverse aorta constriction (TAC) surgery, introducing a 27G stenosis as previously described [26,58]. Briefly, mice were anesthetized, intubated and ventilated (2% isoflurane in air, 50 mL/min; rodent MiniVent, Harvard Apparatus, Germany) on a heated operation table. The thoracic cavity was opened by a median sternotomy until the 3rd rib and a blunted needle (27G) was placed on the aortic arch between the innominate artery and left common carotid. A 7-0 suture was tied onto the needle. Immediately afterwards the needle was removed, and the thoracic cavity was closed. Sham mice were treated identically except for the ligation of the thoracic aorta. Before and over a period of 3 d (every 12 h) after surgery, the animals were treated with analgesic medication (Buprenorphine 0.02 mg/kg s.c.) and monitored daily until the end of the study. Directly after surgery mice were randomized to receive either AAV9-Luc or AAV9-Tfeb by an investigator blinded to treatment and analyses. 10^12^ vector genomes of AAV9 were injected into the tail vein using a 30G needle to overexpress TFEB in cardiomyocytes (AAV9-Tfeb) of Sham (*n* = 9) and TAC (*n* = 14) mice. AAV9-Luc was used as a control in Sham (*n* = 9) and TAC (*n* = 9) mice. The effectiveness of TAC was confirmed by performing transthoracic echocardiography at baseline and after 4 weeks. Potential confounders such as the order of treatments and measurements, as well as animal/cage location, were minimized by prespecified and consecutive handling of the animals. Mice were sacrificed by cervical dislocation and removal of the heart after 4 weeks of surgery and directly after transthoracic echocardiography while still being in isoflurane anesthesia. At this time point, the integrity of aortic banding was confirmed by inspection of the surgical constriction. Hearts, lungs, and livers were harvested; body weight and organ weights were measured and normalized to tibia length.

### 4.3. Transthoracic Echocardiography

Two-dimensional transthoracic echocardiography was performed as previously described [6,7,8,26,59]. Mice were anesthetized with 2% isoflurane and kept warm on a heated platform. Temperature and electrocardiography were continuously monitored. Systolic cardiac function and morphology were assessed with a VisualSonics Vevo 2100 High-Resolution Imaging System with the use of a high-resolution (38 MHz) transducer. The examiner was blinded for the treatment groups. The following parameters were measured: thickness of the left ventricular posterior wall (PWths, PWthd); septum (IVSths, IVSthd) at systole (s); diastole (d); left ventricular ejection fraction (LVEF); stroke volume (SV); left ventricular enddiastolic (LVEDD) and endsystolic (LVESD) dimensions. The left ventricular fractional shortening (FS) was calculated as following FS = [(LVEDD − LVESD)/LVEDD] × 100. Diastolic parameters were evaluated as follows: Doppler flow profiles were acquired using pulsed-wave Doppler in the apical 4-chamber view. The sample volume was placed close to the tip of the mitral leaflets in the mitral orifice parallel to the blood flow to record maximal transmitral flow velocities. A simultaneous mitral inflow and aortic outflow profile was recorded, which allows the measurement of the isovolumetric relaxation time (IVRT, time interval between aortic valve closure and mitral valve opening).

### 4.4. Histological Analyses

The apex of the harvested hearts was embedded in gum tragacanth (Sigma-Aldrich Chemie GmbH, Germany) and then stepwise frozen in isopentane and liquid nitrogen. Embedded tissue was stored at −80 °C until further usage. A cryotome (Leica CM 3050 S, Leica Microsystems GmbH, Wetzlar, Germany) was used for sections that were stained with haematoxylin and eosin, and Picrosirius Red as previously described [6,8,26,59]. For Picrosirius Red stain, 6µm thick myocardial sections were fixed in ice-cold acetone for 10 min and afterwards washed twice with 98% ethanol for 5 min. Slides were then stained for 30 min in Picrosirius Red F3BA (Polysciences, Inc., Warrington, PA, USA). After rinsing with distilled water, slides were washed with 98% and absolute ethanol, dehydrated with xylol and mounted with Vitro-Clud (R. Langenbrinck GmbH, Emmendingen, Germany). Images were acquired using a Leica CTR 6500 HS microscope (Leica Microsystems GmbH, Wetzlar, Germany). Fibrosis was quantified using ImageJ software 1.42c (http://rsb.info.nih.gov/ij, last access 29 April 2022). In tile scan images, the borders of the myocardium were selected manually, and staining artefacts were removed. Using a predefined threshold, the red area representing fibrosis was measured and the results are presented as a percent of the total area.

### 4.5. RNA Isolation, cDNA Synthesis and Quantitative Real-Time PCR

Total RNA was isolated from the interventricular septum using TRIzol^®^ reagent (Invitrogen™, Life Technologies Corporation, Carlsbad, CA, USA) and the FastPrep-24™ instrument (MP Biomedicals, Santa Ana, CA, USA) following the manufacturer’s protocol and as recently reported [8,20,59,60]. The SuperScript^®^ First-Strand Synthesis System (Invitrogen™, Life Technologies Corporation, Carlsbad, CA, USA) was used to synthesize 1.5 µg RNA per sample in accordance with the manufacturer’s instructions. Quantitative real-time polymerase chain reaction (qRT-PCR) was performed using Power SYBR^®^ Green PCR Master Mix (Applied Biosystems) and self-designed primers (for primer sequences see Appendix A) on a Step-One™ Plus thermocycler (Applied Biosystems, Waltham, MA, USA) using the standard curve method as described recently [8,58,61,62]. The expression of individual genes was normalized to the geometric mean of the three stably expressed internal control genes *mitochondrial ribosomal protein L13* (*Mrpl13*), *importin 8* (*Ipo8*), and *phosphoglycerate kinase 1* (*Pgk1*) selected from obtained RNA sequencing data and according to previously published work [63].

### 4.6. RNA-Sequencing

Total RNA analyses were evaluated by an Agilent 2100 Bioanalyzer (Agilent Technologies, Inc., Santa Clara, CA, USA). Library preparation of 500 ng RNA was performed using the Illumina TruSeq Stranded mRNA Kit. To check integrity, DNA was evaluated again using an Agilent 2100 Bioanalyzer. The initial quality check of RNA-sequencing results was done by FASTQC software (v0.11.5; Babraham Bioinformatics, Babraham, UK; available online at www.bioinformatics.babraham.ac.uk/projects/fastqc, last access 29 April 2022) [64]. Sequencing reads were mapped to the mouse whole genome (mm10) using STAR aligner (v 2.5.3a; default parameters) [65]. Read counts for each gene (Gencode vM12) were extracted from the BAM file using featureCounts software (v1.5.1) [66]. In order to avoid background signal noise, genes with less than 10 reads over all samples were excluded. Read counts from different biological groups were subjected to differential expression analysis using the DESeq2 R statistical package (v1.16.1) [67]. Genes with adjusted *p*-value (Benjamini–Hochberg procedure) lower than 0.05 were considered significantly differentially expressed in the respective comparison. Pathway enrichment analysis was performed using DAVID (Database for Annotation, Visualization, and Integrated Discovery) Bioinformatics Resources 6.8 (https://david.ncifcrf.gov, last access 29 April 2022) using the outputs BP_ALL (all biological process terms) and MF_ALL (all molecular function terms). The transcriptome data can be found under EBI Annotare v.2.0 (Project-ID: E-MTAB-9585).

### 4.7. Protein Extraction

The FastPrep-24™ instrument (MP Biomedicals, Santa Ana, CA, USA) was used to homogenize tissue in radioimmunoprecipitation assay (RIPA) buffer (50 mM Tris-HCl, pH 7.5, 150 mM sodium chloride, 1 mM EDTA, 1% (*v*/*v*) Nonidet P-40, 0,1% (*w*/*v*) sodium dodecyl sulfate (SDS), 0.5% (*w*/*v*) sodium deoxycholate) adjusted to pH 8.0 and supplemented with protease inhibitors (cOmplete, Roche Diagnostics GmbH, Rotkreuz, Switzerland) and phosphatase inhibitors (PhosphoStop, Roche Diagnostics GmbH, Rotkreuz, Switzerland). Micro Packaging Vials with 2.8 mm Precellys ceramic beads (PEQLAB Biotechnology GmbH, Erlangen, Germany) were used for homogenization. Lysates were cleared by centrifugation at 15,000× *g* for 15 min at 4 °C and protein content was quantified using Pierce^®^ BCA reagent (Thermo Fischer Scientific Inc., Waltham, MA, USA). After adding 6 × Laemmli buffer (300 mM Tris-HCl, pH 6.8; 12% (*w*/*v*) SDS; 0.1% (*w*/*v*) bromophenol blue, 50% (*v*/*v*) glycerol and 1/6 β-mercaptoethanol added before usage), samples were heated for 5 min at 95 °C and afterwards resolved by SDS-PAGE and transferred onto nitrocellulose or polyvinylidene difluoride (PVDF) membranes (GE healthcare, Chicago, IL, USA). Membranes were blocked with 5% skim milk powder or 5% bovine serum albumin (BSA) in TBS-T (20 mM Tris, 150 mM NaCl, 0.1% Tween 20; pH 7.6) for 1 h. For immunoblotting following antibodies were used for overnight incubation at 4 °C: monoclonal anti-β/slow myosin heavy chain (MyHC) (NOQ7.5.4D, mouse, 1:1000, Sigma-Aldrich Chemie GmbH, Darmstadt, Germany), anti-p62 (guinea pig, 1:1000, Santa Cruz Biotechnology, Inc., Dallas, TX, USA), anti-microtubule-associated protein 1A/1B light chain 3B (anti-LC3B; detecting LC3B-I and LC3B-II, rabbit, 1:750, Cell Signalling Technology Inc., Danvers, MA, USA), anti-TFEB (rabbit, 1:1000, Bethyl Laboratories Inc., Montgomery, TX, USA) and anti-Beclin-1 (rabbit, 1:1000, Cell Signalling Technology Inc., Danvers, MA, USA). Loading was controlled with anti-glyceraldehyde-3-phosphate dehydrogenase (clone 6C5, mouse, 1:10,000, Merck Millipore, Billerica, MA, USA). Horseradish Peroxidase (HRP)-linked IgG goat anti-mouse, goat anti-rabbit (both 1:3000, Cell Signalling Technology Inc., Danvers, MA, USA) or goat anti-guinea pig (1:3000, Abcam, Cambridge, UK) were used as secondary antibodies. Proteins were visualized with a chemiluminescence system (SuperSignal^®^ West Pico Chemiluminescent substrate, Thermo Fischer Scientific Inc., Waltham, MA, USA) according to the manufacturer’s protocol.

### 4.8. Statistical Tests

Differences between the two groups were evaluated with an unpaired two-tailed Student’s *t*-test. One-way analysis of variance (ANOVA) followed by Tukey post hoc test was used for comparison of more than two independent groups. Weight analysis and echocardiography data are presented as box-and-whisker plots with boxes showing the mean and interquartile range (IQR) and whiskers indicating the minimum and maximum values. All qRT-PCR gene expression data are shown as mean ± standard error of the mean (SEM) in bar plots. Plots and statistics were performed by using GraphPad Prism^®^ 8 program (GraphPad Software, San Diego, CA, USA; version 8.03) and Adobe Illustrator CS6 (Adobe Inc. Mountain View, CA, USA), version 16.0.0.

## 5. Conclusions

Cardiomyocyte-specific TFEB overexpression sensitizes the heart to pressure-overload-induced cardiac stress resulting in heart failure. This phenotype can be attributed to TFEB-induced ALP activation. The treatment of ALP-related diseases such as Pompe’s disease by TFEB gene therapy warrants careful consideration and specific tissue-targeting that excludes the heart.

## Figures and Tables

**Figure 1 ijms-23-05943-f001:**
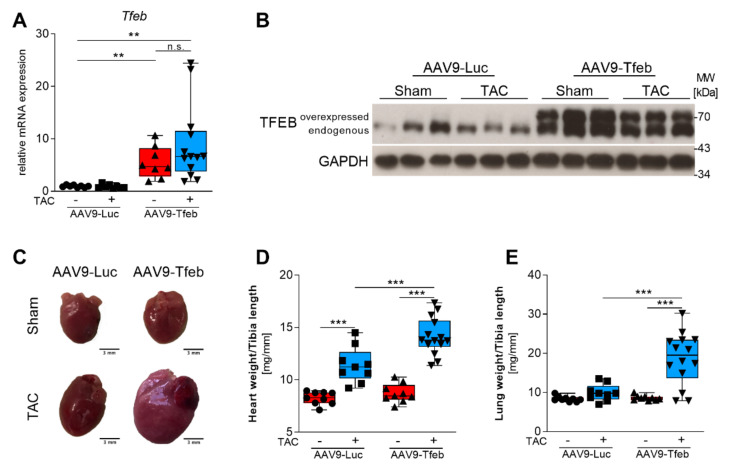
Cardiomyocyte-specific overexpression of TFEB leads to heart failure in response to 28 days of pressure overload. (**A**) qRT-PCR results of Tfeb mRNA expression normalized to geometric mean of Ipo8, Mrpl13 and Pgk1 and shown as mean ± SEM. ** *p* < 0.01; n.s. not significant. (**B**) Western blot analysis of protein lysates from hearts to analyse TFEB protein levels. Overexpressed and endogenous TFEB are indicated. Glyceraldehyde-3-phosphate dehydrogenase (GAPDH) protein content was used as loading control. (**C**) Gross morphology of representative hearts of sham and TAC treated AAV9-Luc and AAV9-Tfeb mice are shown. (**D**) Heart weight/tibia length (HW/TL) and (**E**) Lung weight/tibia length (LW/TL) ratios of AAV9-Luc and AAV9-Tfeb mice 28 days after sham and TAC surgery are shown. Box plots show median and interquartile range (IQR) ± minimum to maximum values. *** *p* < 0.001.

**Figure 2 ijms-23-05943-f002:**
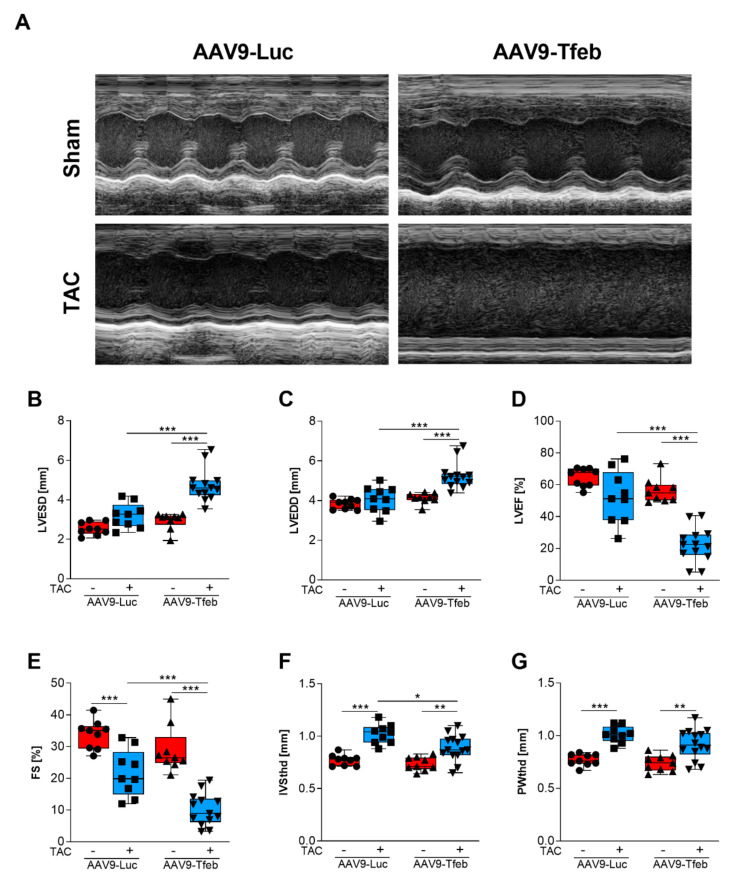
Cardiomyocyte-specific overexpression of TFEB leads to heart failure in response to 28 days of TAC. (**A**) Representative echocardiographic M-mode tracings of sham and TAC treated AAV9-Luc and AAV9-Tfeb mice. Echocardiographic assessment of left ventricular endsystolic (LVESD, (**B**)) and enddiastolic diameter (LVEDD, (**C**)), left ventricular ejection fraction (LVEF, (**D**)), fractional shortening (FS, (**E**)), thickness of the interventricular septum in diastole (IVSthd, (**F**)) and posterior wall thickness in diastole (PWthd, (**G**)) of sham and TAC treated AAV9-Luc and AAV9-Tfeb mice after 28 days. Boxes show IQR ± minimum to maximum values. * *p* < 0.05; ** *p* < 0.01; *** *p* < 0.001.

**Figure 3 ijms-23-05943-f003:**
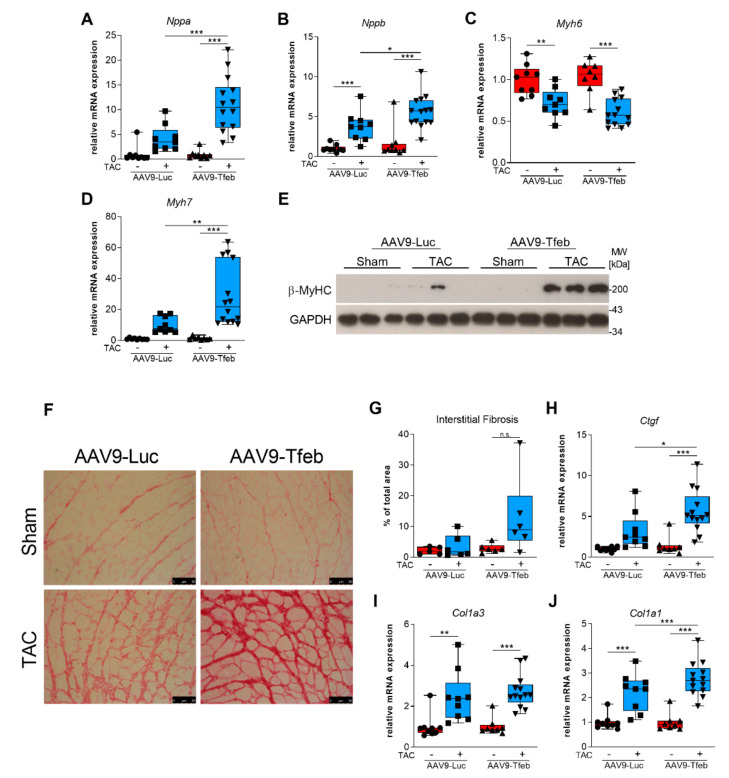
Overexpression of TFEB aggravates pressure-overload-induced cardiac stress response and interstitial fibrosis. qRT-PCR of RNAs isolated from hearts of AAV9-Luc and AAV9-Tfeb mice 28 days after sham and TAC surgery. Gene expression of *Nppa* ((**A**), atrial natriuretic factor), *Nppb* ((**B**), B-type natriuretic peptide, *Myh6* ((**C**), α-myosin heavy chain), and *Myh7* ((**D**), β-myosin heavy chain), is shown. Data were normalized to geometric mean of *Mrpl13*, *Ipo8* and *Pgk1* and presented as mean ± SEM. * *p* < 0.05; ** *p* < 0.01; *** *p* < 0.001. (**E**) Western Blot analysis of β-MyHC (β-myosin heavy chain). Glyceraldehyde-3-phosphate dehydrogenase (GAPDH) protein content was used as loading control. (**F**) Representative Picrosirius Red staining of histological sections from hearts of sham and TAC operated AAV9-Luc and AAV9-Tfeb mice to identify interstitial fibrosis. Scale bars: 50 µm. (**G**) Interstitial fibrosis is presented as percentage of total area of the histological cross section (*n* = 5–6 per group). (**H**–**J** Quantification of *connective tissue growth factor* ((**H**), *Ctgf*), *collagen alpha-1 type III* ((**I**), *Col III*), and *collagen alpha-1 type I* ((**J**), *Col I*) expression by qRT-PCR normalized to geometric mean of *Ipo8*, *Mrpl13* and *Pgk1*. Data are shown as mean ± SEM. * *p* < 0.05; *** *p* < 0.001.

**Figure 4 ijms-23-05943-f004:**
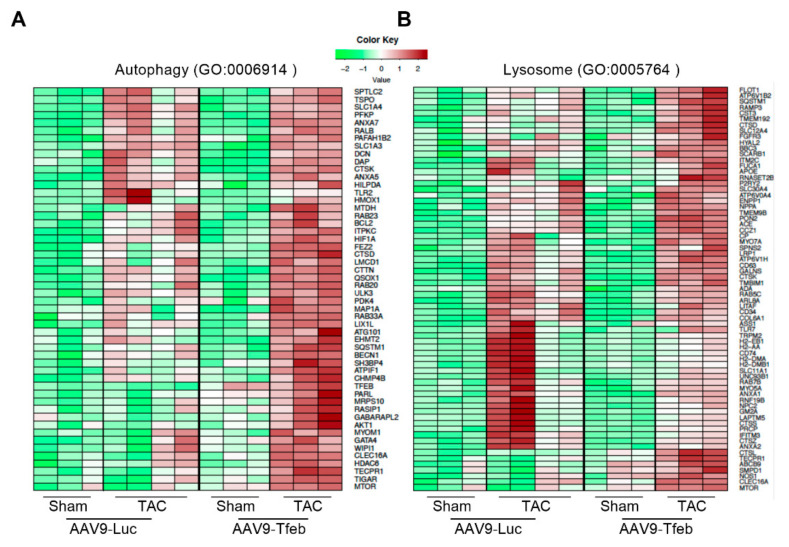
RNA sequencing reveals that TFEB increases the expression of ALP genes during TAC. Heat maps showing the expression of genes that are enriched GO-terms (**A**) autophagy (GO:0006014) and (**B**) lysosome (GO:0005764).

**Figure 5 ijms-23-05943-f005:**
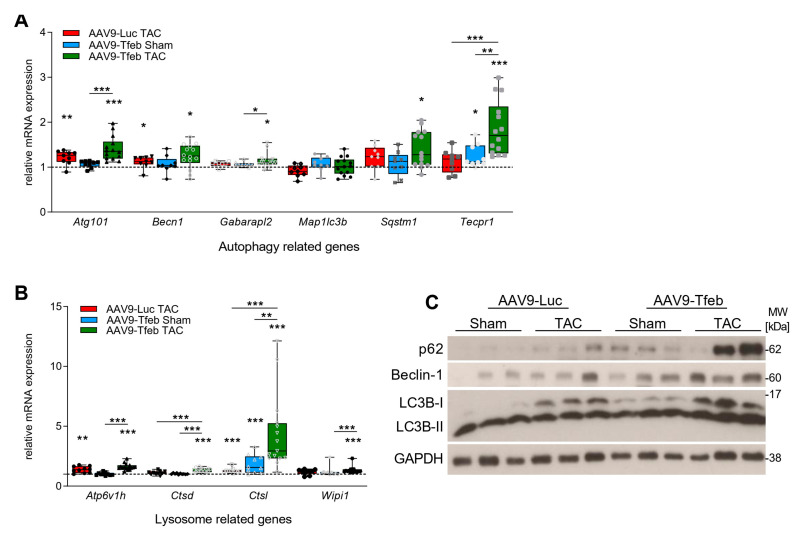
Validation of selected genes enriched in the GO-term analysis by qRT-PCR and Western blot analysis. (**A**,**B**) qRT-PCR to quantify gene expression of autophagy related genes (**A**) of *autophagy-related 101* (*Atg101*), *Beclin-1* (*Becn1*), *gamma-aminobutyric acid A receptor-associated protein-like 2* (*Gabarapl2*), *microtubule-associated proteins 1A/1B light chain 3B* (*Map1lc3b*), *sequestosome 1* (*Sqstm1*/p62), *tectonin beta-propeller repeat containing 1* (*Tecpr1*), and lysosome related genes (**B**) *ATPase, H+ transporting, lysosomal V1 subunit H* (*Atp6v1h*), *cathepsin D* (*Ctsd*), *cathepsin L* (*Ctsl*) and *WD repeat domain phosphoinositide-interacting protein 1* (*Wipi1*) in the interventricular septum from sham (AAV9-Luc, AAV9-Tfeb: *n* = 9) and TAC (AAV9-Luc: *n* = 9, AAV9-Tfeb: *n* = 14) operated mice. Genes involved in autophagy (GO:0006014) and lysosome (GO:0005764) are separately shown. Data were normalized to geometric mean of *Mrpl13*, *Ipo8* and *Pgk1* and presented as mean ± SEM. Values indicate relative expression levels related to the AAV9-Luc sham group, which group mean was set to 1 (±SEM). * *p* < 0.05; ** *p* < 0.01; *** *p* < 0.001. (**C**) Western blot of proteins from left ventricular tissue lysates of sham (AAV9-Luc and AAV9-Tfeb) and TAC (AAV9-Luc and AAV9-Tfeb) treated mice (*n* = 3 each) using anti-p62, ant-beclin-1 and anti-LC3 antibody, as indicated. Glyceraldehyde-3-phosphate dehydrogenase (GAPDH) protein content was used as the loading control.

## Data Availability

The datasets during and/or analysed during the current study available from the corresponding author on reasonable request. The transcriptome data can be found under EBI Annotare v.2.0 (Project-ID: E-MTAB-9585). All other data are available from the corresponding author upon reasonable request.

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
