# Peer review of "The Transcription Factor EB (TFEB) Sensitizes the Heart to Chronic Pressure Overload"

_ijms, 2022, doi:10.3390/ijms23115943_

Round 1

Reviewer 1 Report

The manuscript by Wundersitz et al. evaluate the consequence of the overexpression of TFEB in adult hear by AAV vectors, under the control of  the cardiomyocyte specific MLC v2.1 promoter, in a well-controlled TAC model. The AAV vectors were administered by tail vein.

Results were compatible with a direct co-relation between levels of TFEB and sensitization of the heart to chronic pressure-overload, resulting in left ventricular dilatation, reduced left ventricular function, diminished hypertrophic response, and diastolic dysfunction with a restrictive filling pattern. Cardiomyocyte-specific TFEB overexpression also promote TAC-induced interstitial fibrosis and ALP gene expression, associated with the occurrence of heart failure in response to pressure-overload, and based on these results  authors call to caution to the considered treatment of some lysosome storage diseases (like Pompe).   

The manuscript describes an interesting study, well-controlled and well-executed. The manuscript is clearly written and art work is quite elegant.  

Main points

The most clear previous reference for this study is that by Gatto et al. where the authors evaluate the impact of the administration (systemic delivery) of AAV-Tfeb in a murine model of Pompe disease. They analyze the impact on muscle and heart (in the medium and long-term). Perhaps in the discussion section, these antecedents could be compared more thoroughly, in technical terms (Gatto et al. use the MCK promoter) and functional implications/differences in the present TC model. In the same sense, it could be interesting the discussion on the most appropriate (intramyocardial vs i.v.), especially in the context of a large animal intervention or even the eventual application to humans

Minor points

  • It could be practical to include the time of analysis in some legends
  • Figure 2A is not referred in the text.
  • Figure 5C. In the p62 line (AAV9-Tfeb; TAC) one of the samples included is completely negative in sharp contrast with the other two samples. If this variability is correct, probably it could discussed in the text.

Reviewer 2 Report

Wundersitz et al. investigated the effect of the transcription factor EB in TAC-induced heart failure in mice. Their main findings are that TFEB worsened the macroscopic, microscopic, and molecular signs of heart failure in the TAC animals and these effects are associated with the induction of the autophagy and lysosomal pathway. The experiments seem to be carefully planned and executed. The authors used plenty of state-of-the-art methods. The text is easy to read and the Figures and Tables properly support the message of the text. This reviewer has only some minor comments:

  1. In the legend of Figure 3, fibronectin results are mentioned, however, there is no panel K in Figure 3. Please show the fibronectin results also.
  2. Methods section: it would be helpful if the authors would write the full name of the abbreviation IVRT, i.e., isovolumic relaxation time (beyond its definition).
  3. Discussion: is there any literature data on sex-based differences in the effects of TFEB-induced autophagy and lysosomal pathway in heart failure?
